

# Arctic warming induced by the Laurentide ice sheet topography

Johan Liakka[1] and Marcus Lofverstrom[2]

[1]Nansen Environmental and Remote Sensing Center, Bjerknes Centre for Climate Research, Thormøhlensgate 47, Bergen 5006, Norway
[2]National Center for Atmospheric Research, 3090 Center Green Dr., 80301, Boulder, Colorado, USA

**Correspondence:** johan.liakka@nersc.no

**Abstract.** It is well known that ice sheet-climate feedbacks are essential for realistically simulating the spatio-temporal evolution of continental ice sheet over glacial-interglacial cycles. However, many of these feedbacks are dependent on the ice sheet thickness, which is poorly constrained by proxy data records. For example, height estimates of the Laurentide Ice Sheet (LIS) topography at the Last Glacial Maximum (LGM; ∼21,000 years ago) vary by more than 1 km between different ice-sheet
reconstructions. In order to better constrain the LIS elevation it is therefore important to understand how the mean climate is influenced by elevation discrepancies of this magnitude. Here we use an atmospheric model coupled to a slab-ocean model to analyze the LGM surface temperature response to a broad range of LIS elevations (from 0 to over 4 km). We find that raising the LIS topography induces a widespread surface warming in the Arctic region, amounting to approximately 1.5°C per km elevation increase, or about 6.5°C for the highest LIS. The warming is attributed to an increased northward energy flux by
atmospheric stationary waves, reinforced by surface albedo and water vapor feedbacks, which account for about two-thirds of the total temperature response. These results suggest a positive feedback between continental-scale ice sheets and the Arctic temperatures that may help constrain LIS elevation estimates for the LGM and explain differences in ice distribution between the LGM and earlier glacial periods.

# 1 Introduction

The Last Glacial Maximum (LGM), ∼21,000 years before present (21 kyrs BP), was the apex of the last glacial period when the global ice volume reached its maximum value (Peltier and Fairbanks, 2006; Lambeck et al., 2014). In comparison with earlier glacial cycles, the LGM climate conditions are relatively well documented by proxy data records. In addition to the well-established records of Earth's orbital configuration (Berger and Loutre, 2004), atmospheric greenhouse gas concentrations
(Petit et al., 1999; Spahni et al., 2005), and sea surface temperatures (Margo Project Members et al., 2009), the LGM is the only glacial period for which the margins of the North American and Eurasian ice sheets – the Laurentide Ice Sheet (LIS) and the Fennoscandian Ice Sheet (FIS), respectively – can be reliably reconstructed from geological and geomorphological records. These records reveal that the LIS was by far the larger of the two, covering most of the North American continent poleward of



~40°N, while the FIS was comparatively small and mostly confined to northern Europe (Clark and Mix, 2002; Svendsen et al., 2004; Kleman et al., 2013). However, while the horizontal margins of the LGM ice sheets are known, their thickness and vertical extents remain uncertain. This uncertainty is perhaps best reflected in elevation estimates of the LIS, which vary by more than 1 km between contemporary reconstructions (e.g. Peltier, 2004; Kleman et al., 2013; Abe-Ouchi et al., 2015).

As a result, the Paleoclimate Modeling Intercomparison Project (PMIP4; Kageyama et al., 2017) now encourages sensitivity experiments with three distinct ice sheet reconstructions that differ in height by several hundred meters (Kageyama et al., 2017).

Modeling studies have revealed that the continental ice sheets – in particular the LIS due to its larger size and location in the westerly mean flow – had a substantial impact on the atmospheric and oceanic circulation during the last glacial cycle.

For example, it has been shown that the LIS topography may have altered both the strength and orientation of the mid-latitude Atlantic jet stream (Li and Battisti, 2008; Löfverström et al., 2014, 2016; Löfverström and Lora, 2017) and the associated storm tracks and precipitation patterns (Kageyama and Valdes, 2000; Löfverström et al., 2014, 2016). Model experiments have also revealed that a higher LIS elevation helps strengthen the Atlantic Overturning Meridional Circulation (AMOC) and wind-driven North Atlantic gyre circulation, which typically results in a poleward shift of the sea-ice edge and increased

temperatures in the subpolar North Atlantic (Justino et al., 2006; Eisenman et al., 2009; Pausata et al., 2011; Zhang et al., 2014; Zhu et al., 2014; Gong et al., 2015; Colleoni et al., 2016b; Klockmann et al., 2016; Gregoire et al., 2017). Moreover, several studies have shown that the LIS topography can significantly alter the atmospheric stationary wave field (i.e. the zonally asymmetric component of the time-mean atmospheric circulation; Cook and Held, 1988; Kageyama and Valdes, 2000; Roe and Lindzen, 2001; Langen and Vinther, 2008; Colleoni et al., 2016b; Liakka et al., 2016; Löfverström et al., 2014, 2016).

The stationary waves can in turn influence the local temperature and precipitation anomalies that are important for the surface mass balance (Lindeman and Oerlemans, 1987; Roe and Lindzen, 2001; Herrington and Poulsen, 2011; Liakka and Nilsson, 2010; Liakka et al., 2011; Liakka, 2012; Löfverström et al., 2015; Liakka et al., 2016; Löfverström and Liakka, 2016, 2018).

There are also some modeling evidence that the LIS topography can influence the meridional (equator-to-pole) temperature profile, especially in the northern high latitudes (Justino et al., 2005; Ullman et al., 2014; Zhang et al., 2014). For example,

Ullman et al. (2014) found that high-end estimates of the LGM LIS elevation increases the mean Arctic surface temperature by several degrees Celsius compared to lower reconstructions. While the authors argued that this response may be attributed to a reduced snow cover in Siberia as a result of changes in the atmospheric stationary wave field, they did not explore this narrative further. This result is however noteworthy, as it shows that the elevation of a mid-latitude topographic barrier can substantially influence the (zonal) mean climate in the high latitudes. This further illuminates a gap in our current understanding

of atmosphere-topography interactions and potential feedbacks between continental-scale ice sheets and the temperature in glacial climates.

Understanding the origin, physics/dynamics, and implications of such feedback could potentially help constrain the range of possible LIS elevations at the LGM, and also explain differences in the ice sheet extent between the LGM and earlier glaciations. For example, the Penultimate Glacial Maximum (PGM; ~140 kyrs BP) had a somewhat reversed ice-volume

distribution compared to the LGM, with a larger ice sheet in Eurasia and a comparatively smaller ice sheet in North America



(Svendsen et al., 2004; Wekerle et al., 2016). There is also evidence of extensive Arctic ice shelves from the PGM; similar evidence have not been found from the LGM when the LIS is thought to have been larger (Niessen et al., 2013; Colleoni et al., 2016a; Jakobsson et al., 2016; Nilsson et al., 2017).

Here we explore feedbacks between the LIS elevation and the LGM surface temperature in a comprehensive atmospheric general circulation model (AGCM) coupled to a slab-ocean model. In agreement with Ullman et al. (2014), we find that raising the LIS topography (from 0 to over 4 km) yields a widespread surface warming in the Arctic, culminating at about 6.5°C for the highest LIS. A thorough analysis of the Arctic energy budget reveals that the LIS-induced warming is primarily explained by an increased meridional energy-flux convergence from atmospheric stationary waves, amplified by positive feedbacks from the surface albedo and atmospheric water vapor.

The paper is organized as follows. Section 2 describes the AGCM and the experimental design and section 3 presents the results, which are further discussed in section 4.

## 2   Model and experiments

We use the National Center for Atmospheric Research Community (NCAR) Atmospheric Model version 3 (CAM3; Collins et al., 2006a) with T85 spectral resolution (∼1.4° horizontal resolution) and 26 hybrid pressure-sigma levels in the vertical. Land processes are treated by the Community Land Model 3 (CLM3; Oleson et al., 2004). The planetary boundary conditions are prescribed as typical LGM conditions: the orbital parameters are set to appropriate values for 21 kyrs BP (Berger and Loutre, 2004), and the concentrations of $CO_2$, $CH_4$, and $N_2O$ are prescribed as 185 ppmv (parts per million by volume), 350 ppbv (parts per billion by volume), and 200 ppbv, respectively (Petit et al., 1999; Spahni et al., 2005); CFCs are set to zero. Aerosols, vegetation, and non-glaciated land areas attain their pre-industrial (PI) configuration. We use a standard PI simulation as reference climate, which has been evaluated against observations in e.g. Löfverström et al. (2014) and Liakka et al. (2016) – the model captures the amplitude and spatial variations of the observed climatology (e.g. surface temperature, precipitation, and geopotential height anomalies) to a high degree.

We couple the atmospheric model to a computationally efficient slab-ocean (mixed-layer) model in order to facilitate a high number of experiments. Although the ocean representation is motionless and therefore does not account for changes in circulation, it retains the thermodynamic feedback between the ocean and the atmosphere. Sea-surface temperatures (SSTs) and sea ice are explicitly calculated from the energy balance in the ocean mixed layer (Collins et al., 2006b; Bitz et al., 2012), where the monthly oceanic heat flux convergence field (internal energy source in the mixed layer that represents horizontal energy transport by ocean currents) and annual mixed-layer depth are derived from a 50 year LGM timeslice from the fully coupled TraCE-21ka (Transient Climate Evolution over the last 21,000 years; Liu et al., 2009; He, 2011). All simulations are integrated for 60 years, of which the first 35 model years are regarded as spin-up, and the remaining 25 years are averaged to create the atmospheric climatological fields used in the analysis.

The ice sheets in North America and Eurasia are derived from the LGM reconstruction in Kleman et al. (2013), which is similar to other contemporary reconstructions (cf. black dashed contours in Fig. 1 with  Peltier, 2004; Abe-Ouchi et al., 2015);



the maximum LIS elevation is approximately 3.3 km. As the objective of this study is to evaluate the importance of the LIS topography on the surface temperature, the height of the FIS remains constant (Kleman et al., 2013) in all experiments.

We conduct a total of six steady-state simulations with different heights of the LIS. In each simulation the LIS elevation is multiplied by a uniform constant $N$, which takes on values between 0 and 1.25 in increments of 0.25. The LIS morphology and spatial extent therefore remains the same in all experiments, but the elevation is altered: $N = 0$ represents present-day North American orography and LGM land albedo (glacial mask), $N = 1$ is the "standard" LGM case with unscaled LIS topography (~3.3 km maximum elevation; Kleman et al., 2013), and $N = 1.25$ has a maximum LIS elevation of ~4.1 km, approximately similar to the ICE-5G reconstruction (Peltier, 2004). The LGM sensitivity experiments are referred to as LIStopo$N$, where $N$ is the topography scaling factor.

## 3 Results

### 3.1 Surface temperature response

Figure 1a,b shows the annual-mean surface temperature from the PI and LIStopo0 simulations. The LIStopo0 surface temperature is substantially colder than PI, particularly in high latitudes. This cooling is explained by the high surface albedo from the ice sheets, a more extensive sea ice cover, and lower concentrations of atmospheric greenhouse gases.

The annual-mean surface temperature response to the LIS topography is shown in Fig. 1c-g. This is evaluated as the difference in surface temperature with respect to the LIStopo0 simulation. The largest cooling is confined to the LIS area (as a result of the surface elevation change), and some smaller cold anomalies are found in the mid and subpolar North Atlantic. The cooling east of the LIS topography is a typical downstream response to topographically-induced stationary waves as cold air is advected from the ice sheet interior by the westerly mean flow (e.g. Roe and Lindzen, 2001; Liakka et al., 2011). Elsewhere, raising the LIS topography results in warmer temperatures, particularly in non-glaciated high-latitude land areas and in the Arctic basin (Fig. 1c-g).

The annual-mean surface temperature in the Arctic region increases by approximately 1.5°C per km of LIS elevation (Fig. 2). The high-latitude warming is present in all seasons (Fig. 2), but it is strongest in boreal winter (December-January-February; DJF). The seasonal persistence of the Arctic warming suggests that it is driven by changes in both atmospheric dynamics and physics. While the effect of dynamics is typically more pronounced in the winter season, many radiative features (such as the surface albedo-temperature feedback) are more important in summer when the insolation is higher (insolation in boreal winter is negligible at these latitudes).

### 3.2 Role of atmospheric dynamics

Motivated by the apparent connection between the LIS height and Arctic temperatures, the following sections analyze how atmospheric meridional heat flux changes when raising the LIS topography.





### 3.2.1 Basic theory

In steady-state, changes in the atmospheric energy storage is zero for annual climatologies (Peixoto and Oort, 1992; Trenberth et al., 2001; Serreze et al., 2007), implying that the atmospheric energy balance can be written as (e.g Serreze et al., 2007):

$$C = S - R, \tag{1}$$

where $C \equiv -\nabla \cdot \boldsymbol{F}$ is the convergence of vertically-integrated (annual mean) horizontal energy flux, $R$ the net radiation at the top-of-the-atmosphere (TOA), and $S$ the energy balance at the surface ($R$ and $S$ are both positive downward). The surface energy balance is defined as the sum of the net radiation and turbulent fluxes. Similarly, the latent energy-flux convergence ($C_L \equiv -\nabla \cdot \boldsymbol{F_L}$) is proportional to the difference between precipitation ($P$) and evaporation ($E$):

$$C_L = L_v(P - E), \tag{2}$$

with the latent heat of evaporation $L_v = 2.5 \times 10^6$ J kg$^{-1}$. The dry-static energy-flux convergence ($C_{DS}$) is defined as the residual of the total and latent energy fluxes[1] ($C - C_L$), i.e.:

$$C_{DS} = S - R - L_v(P - E). \tag{3}$$

The implied (zonally and vertically integrated) northward energy flux at each latitude is obtained by integrating Eq. 1:

$$F(\phi) = -a^2 \int_0^{2\pi} \int_{-\pi/2}^{\phi} C(\phi', \lambda') \cos(\phi') d\phi' d\lambda', \tag{4}$$

where $a$ is Earth's radius, $\lambda$ is the longitude, and $\phi$ the latitude (both defined in radians). The equivalent northward fluxes of latent ($F_L$) and dry-static ($F_{DS}$) energy are obtained by substituting $C$ with $C_L$ and $C_{DS}$ in Eq. 4.

The energy flux quantities can be further decomposed into the relative contributions from the zonal-mean circulation, stationary eddies, and transient eddies by using atmospheric state variables at model levels; see Peixoto and Oort (1992) and Appendix A for details.

### 3.2.2 Meridional flux of atmospheric energy

Figure 3a-c shows the implied atmospheric northward energy fluxes ($F$, $F_L$, and $F_{DS}$) from our simulations. There is a slight increase of total energy flux ($F$) in the LGM simulations with respect to PI, and the Northern Hemisphere (NH) peak value is shifted slightly equatorward (Fig. 3a). The LGM change in energy flux is largely represented by a comparable shift in dry static energy ($F_{DS}$) (Fig. 3c), while the latent energy flux ($F_L$) shows an overall reduction in the NH mid-latitudes (Fig. 3b). The increase of total energy transport at the LGM (with respect to PI) is in agreement with previous results from both coupled atmosphere-ocean models, and from atmosphere models forced by prescribed sea-surface conditions (Hall et al., 1996;

---

[1]The kinetic energy is neglected as it is much smaller (typically two orders of magnitude) than the dry-static and latent energy contributions.





Hewitt et al., 2003; Shin et al., 2003; Li and Battisti, 2008; Murakami et al., 2008). Similarly, the increase (decrease) of dry-static (latent) energy flux is a typical LGM response in fully-coupled models (Li and Battisti, 2008; Murakami et al., 2008) and is partially explained by a weaker hydrological cycle in colder climates (Alexeev et al., 2005; Held and Soden, 2006).

To investigate the role of the LIS topography in more detail, we analyze how different atmospheric circulation regimes influence the meridional energy flux. Figure 3d-f shows how the time- and zonal-mean atmospheric circulation ($F_{DSM}$), stationary eddies ($F_{DSS}$), and transient eddies ($F_{DST}$) contribute to the meridional flux of dry-static energy (see Appendix A for details).

The majority of the increase in low-latitude dry-static energy flux is attributed to changes in the mean circulation (Fig. 3d). These changes are however not directly attributed to the LIS topography, as all LGM simulations show a similar response. The LIS topography is found to be more important for the meridional dry-static energy flux from stationary eddies ($F_{DSS}$) and transient eddies ($F_{DST}$). In LIStopo0, $F_{DSS}$ is roughly similar to the PI, whereas the peak $F_{DST}$ is somewhat higher and shifted equatorward (Fig. 3e,f). Raising the LIS elevation yields a gradual increase (decrease) of stationary (transient) dry-static energy flux in the NH extratropics (Fig. 3e,f), in broad agreement with the fully coupled simulations in Li and Battisti (2008) and Murakami et al. (2008). Here we demonstrate that these changes can be primarily attributed to the LIS topography, as all other boundary conditions remain unchanged in our LGM sensitivity simulations.

### 3.2.3 Energy-flux convergence in the Arctic

While the meridional energy flux (as calculated by Eq. 4) is valuable for identifying structural changes of the large-scale atmospheric circulation, it reveals limited information on how the mean climate responds. For that purpose, it is instead more useful to consider the energy-flux convergence (meridional derivative) rather than the flux itself (see Eq. 1).

Table 1 shows the horizontal atmospheric energy-flux convergence in the Arctic polar cap (area-weighted average of all grid points poleward of 70°N). The energy-flux convergence in the PI simulation amounts to 102 W m$^{-2}$, which is in close agreement with estimates from atmospheric reanalysis data (100 to 103 W m$^{-2}$; Serreze et al., 2007). With respect to PI, the total energy-flux convergence ($C$) in LIStopo0 is reduced by 6 W m$^{-2}$, which is primarily explained by a decrease in the latent energy-flux convergence ($C_L$) (Table 1). Raising the LIS topography yields a gradual increase of the total energy-flux convergence ($C$) by an average rate of ~1.5 W m$^{-2}$ km$^{-1}$, resulting in similar values to PI for the highest LIS. This increase stems from an enhanced contribution from the dry-static energy-flux convergence ($C_{DS}$), in particular from stationary waves ($C_{DSS}$; Fig. 3 and Table 1); the latent energy-flux convergence ($C_L$) is approximately the same in all LGM simulations. For the highest LIS, the stationary-wave contribution to the total Arctic energy-flux convergence even dominates over the contribution from transient eddies ($C_{DST}$; Table 1). The reduction in transient eddy activity is characteristic for a reduced storminess at the LGM; see Li and Battisti (2008), Donohoe and Battisti (2009) and Rivière et al. (2018) for further discussions on this topic.

### 3.3 Other feedbacks

The results in Fig. 3 and Table 1 demonstrate that the LIS topography has a dominant influence on the LGM stationary-wave field (in agreement with Cook and Held, 1988; Kageyama and Valdes, 2000; Löfverström et al., 2014). Although the



increased energy-flux convergence from the LIS-induced stationary waves is largely compensated by a comparable reduction from transient eddies, the net effect is a positive contribution (warming) to the Arctic energy balance ($\delta C$ in Table 1). All else being equal, we can crudely estimate the influence of the LIS-induced atmospheric circulation on the Arctic temperature by assuming a typical value of the surface temperature (Planck) feedback parameter ($\lambda_T = 3.2$ W m$^{-2}$ K$^{-1}$; Flato et al., 2013).

For the highest LIS, the contribution from the atmospheric circulation is roughly $\delta C/\lambda_T \approx 2°$C, which is significantly lower than the $\sim 6.5°$C seen in Fig. 2. This difference suggests that other (positive) feedbacks are important for the LIS-induced warming as well.

As seen in Eq. 1, changes in the total energy-flux convergence ($C$) reflect an imbalance between the energy fluxes at the surface ($S$) and at the top of the atmosphere ($R$). For a climate in balance, $S$ is close to zero over land and represented by

the horizontal heat flux divergence in the ocean (i.e. $S = -C_{ocean}$; Trenberth et al., 2001; Serreze et al., 2007). As we use a slab-ocean model, the representation of the ocean heat flux is identical in our simulations, implying that the surface energy balance ($S$) over the ocean remains unchanged when changing the LIS elevation. Any LIS-induced change in the atmospheric energy-flux convergence should therefore be compensated by an equivalent change in the TOA net radiation balance:

$$\delta C + \delta R = 0. \qquad (5)$$

The influence of the LIS elevation on the Arctic TOA net radiation is shown in the left-most column of Table 2. As suggested in Eq. 5, these values have a similar magnitude as the atmospheric energy-flux convergence ($\delta C$ in Table 1). The relatively small differences stem from the fact that the (annual-mean) Arctic surface energy balance is not completely identical in all simulations, but varies between 9.1±0.2 W m$^{-2}$ as a result of slow processes in the land model. These small inconsistencies are however not important for the interpretation of our results and conclusions.

To evaluate the contributions from individual feedbacks to the LIS-induced Arctic warming, the TOA net radiation change ($\delta R$) is separated into radiative contributions from changes in surface albedo ($\delta R_\alpha$), water vapor and lapse rate ($\delta R_{wv+lr}$), total cloudiness ($\delta R_{cld}$), and surface temperature (i.e. the Planck feedback: $\delta R_T$). The estimated strengths of these feedbacks are obtained by separately calculating the contributions to the shortwave (SW) and longwave (LW) parts of the radiative spectrum. The SW decomposition is carried out using the Approximative Partial Radiative Perturbation (APRP) method from

Taylor et al. (2007); the LW decomposition is provided in Appendix B. While the surface albedo ($\delta R_\alpha$) and Planck ($\delta R_T$) feedbacks influence the SW and LW radiation separately, the other terms ($\delta R_{wv+lr}$ and $\delta R_{cld}$) are assumed to contribute to both (see Appendix B). Note that the residual term from the APRP method is omitted here as it is not relevant for the discussion. This term is also comparably small so that the total TOA radiation change is approximately equal to the sum of all individual feedbacks, i.e.:

$$\delta C + \delta R_\alpha + \delta R_{wv+lr} + \delta R_{cld} + \delta R_T \approx 0. \qquad (6)$$

The radiative contribution from each feedback is shown in Table 2. It is evident that the surface albedo ($\delta R_\alpha$) and water vapor ($\delta R_{wv+lr}$) are the most important (positive) feedbacks when raising the LIS elevation; cloud feedbacks ($\delta R_{cld}$) are virtually negligible. For the highest LIS elevations, changes in surface albedo and water vapor together yield a positive radiative contribution by about 14 W m$^{-2}$, thus exceeding the contribution from the energy-flux convergence ($\delta C$ in Table 1)



by approximately a factor 2. Of these two feedbacks, the water vapor feedback is overall about twice as large as the surface albedo feedback, mediated by a ∼30% increase of the total precipitable water content between LIStopo0 and LIStopo1.25. To compensate for the warming contributions from the atmospheric energy-flux convergence, and the water vapor and albedo feedbacks, there is an increased outgoing LW radiation from the surface as a result of higher temperatures (Planck feedback; $\delta R_T$), amounting to about 21 W m$^{-2}$ for the highest LIS reconstruction (Table 2).

## 4 Discussion and concluding remarks

Here we investigate how the LIS topography influences the Arctic surface temperature, using a comprehensive AGCM coupled to a slab-ocean model. Our results show that increasing the LIS elevation (from 0 to over 4 km), while keeping all other boundary conditions fixed at their LGM configuration, results in an annual-mean Arctic warming in excess of 6.5°C. This warming is primarily attributed to a net increase in the atmospheric energy-flux convergence in high latitudes, which is further reinforced by positive feedbacks from a reduced surface albedo and a higher atmospheric water vapor content.

The correlation between Arctic temperatures and the LIS elevation suggests that LGM LIS may have helped reduce the equator-to-pole temperature gradient. This is also supported by annual-mean surface mass balance (SMB) estimates (Fig. 4), evaluated as the difference between accumulation (precipitation) and ablation using the Positive-Degree-Day approach (Braithwaite and Olesen, 1989; Reeh, 1991) from the ice-sheet model SICOPOLIS (Greve, 1997; Calov and Greve, 2005). Figure 4 shows that most (non-glaciated) Arctic land areas change from a positive to negative a SMB when raising the LIS elevation, suggesting that the presence of the LIS topography may have helped keeping Alaska and Siberia ice free at the LGM (in agreement with Roe and Lindzen, 2001; Liakka et al., 2016; Löfverström and Liakka, 2016). Furthermore, areas with positive SMB are found in Siberia in all simulations except LIStopo1.25, which suggests that the maximum LIS elevation at the LGM may have been higher than our default LIS reconstruction (LIStopo1; 3.3 km). It is important to stress that this result likely is model dependent, as the LGM mean climate is highly variable among models (e.g. the global-mean surface temperature at the LGM is found to be between 3.1°C and 5.8°C cooler than preindustrial in the PMIP2 models; Braconnot et al., 2007). It is therefore important to assess the impact of LIS elevation on the SMB in other models before we can use this information to constrain the range of possible LIS elevations. With the result presented here, however, we hope to encourage such experiments in the future.

The feedback between continental-scale ice sheets and meridional temperature distribution presented here may also provide a better understanding of glacial environments beyond the LGM. For example, Jakobsson et al. (2016) showed evidence of a thick and partially grounded Arctic ice shelf during the penultimate glacial maximum (PGM), when the LIS is believed to have been significantly smaller than its LGM size (Dyke et al., 2002; Colleoni et al., 2016b; Wekerle et al., 2016). Here we obtain positive SMB across most of the Arctic coastal areas for the lowest LIS topographies ($N \leq 0.5$), while higher LIS elevations yield negative SMB in the same areas (Fig. 4). Hence, these results suggest that the smaller LIS size at the PGM may have been a contributing factor to the formation of an extensive Arctic ice shelf.



The main limitation of this study is that we use a slab-ocean model and thus neglect potential changes in the ocean circulation. However, these changes are not expected to influence the first order conclusions from this study. There are two main reasons for this. First (i), the primary source of the Arctic warming found here, i.e. the increased energy flux by stationary waves, is also featured in many LGM experiments with fully-coupled models (e.g. Li and Battisti, 2008; Murakami et al.,

2008). Second (ii), the direct impact of the ocean circulation on the Arctic temperature is only important in regions that were mostly free of sea-ice at the LGM. Proxy data from the LGM show that essentially only the subpolar North Atlantic sector, a region strongly influenced by the AMOC variability, was characterized by seasonally ice-free conditions poleward of 70°N (Margo Project Members et al., 2009). However, there is strong evidence from several fully-coupled models that raising the LIS elevation yields a stronger AMOC (Justino et al., 2006; Eisenman et al., 2009; Pausata et al., 2011; Ullman et al., 2014;

Zhang et al., 2014; Zhu et al., 2014; Gong et al., 2015; Klockmann et al., 2016; Gregoire et al., 2017), which thus is expected to amplify the Arctic warming signal rather than weaken it.

Put in perspective, the LIS-induced Arctic energy-flux convergence found here ($\sim$6.5 W m$^{-2}$) even exceeds the radiative forcing from doubling the atmospheric $CO_2$ ($\sim$4 W m$^{-2}$; e.g. Hansen et al., 1997), emphasizing the importance of the LIS topography for the LGM stationary wave field. A similar influence on the stationary waves has not been found for the FIS or

15 the smaller (pre- and post-LGM) configurations of LIS (e.g. Eisenman et al., 2009; Liakka et al., 2016; Gregoire et al., 2017). Hence, it is possible that the stationary-wave induced energy flux, and thus also the associated temperature feedback, is only important when the continental ice sheets are sufficiently large to interact with the westerly mean flow (some evidence of this is shown in Löfverström et al., 2014; Löfverström and Lora, 2017). To explore the limits of this feedback with respect to different ice-sheet configurations and atmospheric mean states is beyond the scope of this study, but we hope that results from the PMIP4

experiments (Kageyama et al., 2017) – in particular the sensitivity experiments with different ice-sheet reconstructions – will help illuminate some of these issues.

*Data availability.* The model output files can be obtained from the first author (johan.liakka@nersc.no) upon request.

## Appendix A: Contributions from the circulation to the meridional energy flux

Here we estimate the (zonally and vertically integrated) northward energy flux from the time- and zonal-mean circulation, as

well as stationary and transient eddies. We refer to Peixoto and Oort (1992) for a more comprehensive review of this topic. The total meridional energy flux (Eq. 4) can be expressed in atmospheric state variables as:

$$F(\phi) = 2\pi a \cos(\phi) \int_0^{p_s} ([\overline{vh_{DS}}] + [\overline{vh_L}]) \frac{dp}{g}, \tag{A1}$$

where $p$ is the pressure, $p_s$ the surface pressure, $v$ the meridional wind and $g = 9.8$ m s$^{-2}$ the gravitational acceleration. Here, $h_{DS} \equiv c_p T + gz$ represents the dry-static energy (per unit mass), defined as the sum of the internal and potential energy, where

$T$ is the temperature, $c_p = 1004$ J kg$^{-1}$ K$^{-1}$ the specific heat capacity, and $z$ the geopotential height. The latent energy is given





by $h_L \equiv L_v q$, where $q$ is the specific humidity, and $L_v = 2.5 \times 10^6$ J kg$^{-1}$ is the latent heat of evaporation. Overbars denote time mean and square brackets zonal mean. The dry-static energy flux can be separated into the contributions from different components of the atmospheric circulation as:

$$\overline{[vh_{DS}]} = [\overline{v}][\overline{h_{DS}}] + \overline{[v^* h_{DS}^*]} + \overline{[v' h_{DS}']}, \tag{A2}$$

where the terms on the right-hand side represent the zonal-mean circulation, stationary eddies and transient eddies, respectively; asterisks and primes represent deviations from the zonal and time mean states. The equivalent contributions from each circulation regime to the dry-static energy flux is given by:

$$F_{DSM} = 2\pi a \cos(\phi) \int\limits_0^{p_s} [\overline{v}][\overline{h_{DS}}] \frac{dp}{g}, \tag{A3}$$

$$F_{DSS} = 2\pi a \cos(\phi) \int\limits_0^{p_s} \overline{[v^* h_{DS}^*]} \frac{dp}{g}, \tag{A4}$$

$F_{DST} = F_{DS} - F_{DSM} - F_{DSS}. \tag{A5}$

Hence, the contribution from transient eddies is here determined as the residual of the total, zonal mean, and stationary components of the circulation. The equivalent latent energy flux can be obtained from Eqs. A2 to A5 by substituting $h_{DS}$ with $h_L$.

The corresponding horizontal energy-flux convergence is calculated by differentiating Eqs. A3 to A5 with respect to the

latitude ($\phi$), and dividing by $-2\pi a^2 \cos(\phi)$, i.e.:

$$C_{DSM} \equiv -\frac{1}{2\pi a^2 \cos(\phi)} \frac{\partial F_{DSM}}{\partial \phi} = -a^{-1} \int\limits_0^{p_s} \frac{\partial}{\partial \phi} [\overline{v}][\overline{h_{DS}}] \frac{dp}{g}, \tag{A6}$$

$$C_{DSS} \equiv -\frac{1}{2\pi a^2 \cos(\phi)} \frac{\partial F_{DSS}}{\partial \phi} = -a^{-1} \int\limits_0^{p_s} \frac{\partial}{\partial \phi} \overline{[v^* h_{DS}^*]} \frac{dp}{g}, \tag{A7}$$

$$C_{DST} = C_{DS} - C_{DSM} - C_{DST}, \tag{A8}$$

where $C_{DS}$ is defined in Eq. 3.

## Appendix B:  Disentangling longwave feedbacks on the TOA net radiation balance

To estimate the TOA longwave (LW) contributions from changes in the surface temperature ($\delta R_T^{LW}$), water vapor and lapse rate ($\delta R_{wv+lr}^{LW}$), and clouds ($\delta R_{cld}^{LW}$) between two simulations (subscripts 1 and 0), we use the following equations (positive




flux downward):

$$\delta R_T^{LW} = R_{s,1}^{LW} - R_{s,0}^{LW}, \tag{B1}$$

$$\delta R_{wv+lr}^{LW} = (R_{c,1}^{LW} - R_{s,1}^{LW}) - (R_{c,0}^{LW} - R_{s,0}^{LW}), \tag{B2}$$

$$\delta R_{cld}^{LW} = (R_1^{LW} - R_{c,1}^{LW}) - (R_0^{LW} - R_{c,0}^{LW}).. \tag{B3}$$

The subscripts "s" and "c" represent surface and clear-sky fluxes, respectively. The clear-sky and total fluxes are taken directly from the model output, and the surface fluxes are computed from the surface temperature using Stefan Boltzmann's law for black body radiation (surface flux proportional to the fourth power of temperature). An important property of Eqs. B1 to B3 is that the individual contributions add up to total LW change: $\delta R^{LW} = \delta R_T^{LW} + \delta R_{wv+lr}^{LW} + \delta R_{cld}^{LW} = R_1^{LW} - R_0^{LW}$.

Finally, the combined SW and LW contributions to the TOA net radiation changes are evaluated as:

$$\delta R = \delta R^{SW} + \delta R^{LW}, \tag{B4}$$

$$\delta R_\alpha = \delta R_\alpha^{SW}, \tag{B5}$$

$$\delta R_{wv+lr} = \delta R_{clr}^{SW} + \delta R_{wv+lr}^{LW}, \tag{B6}$$

$$\delta R_{cld} = \delta R_{cld}^{SW} + \delta R_{cld}^{LW}, \tag{B7}$$

$$\delta R_T = \delta R_T^{LW}, \tag{B8}$$

where $\delta R^{SW}$ is the total TOA SW change, and the quantities $\delta R_\alpha^{SW}$, $\delta R_{clr}^{SW}$ and $\delta R_{cld}^{SW}$ represent the TOA SW contributions from the surface albedo, clear-sky atmosphere (mainly due to changes in the SW absorption by water vapor) and clouds derived from the APRP method (Taylor et al., 2007).

*Competing interests.* No competing interests are present.

*Acknowledgements.* We thank Johan Kleman for providing the LGM ice sheet reconstruction. The computational resources for the numerical
simulations were provided by the Centre for Scientific Computing in Frankfurt, Germany.



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





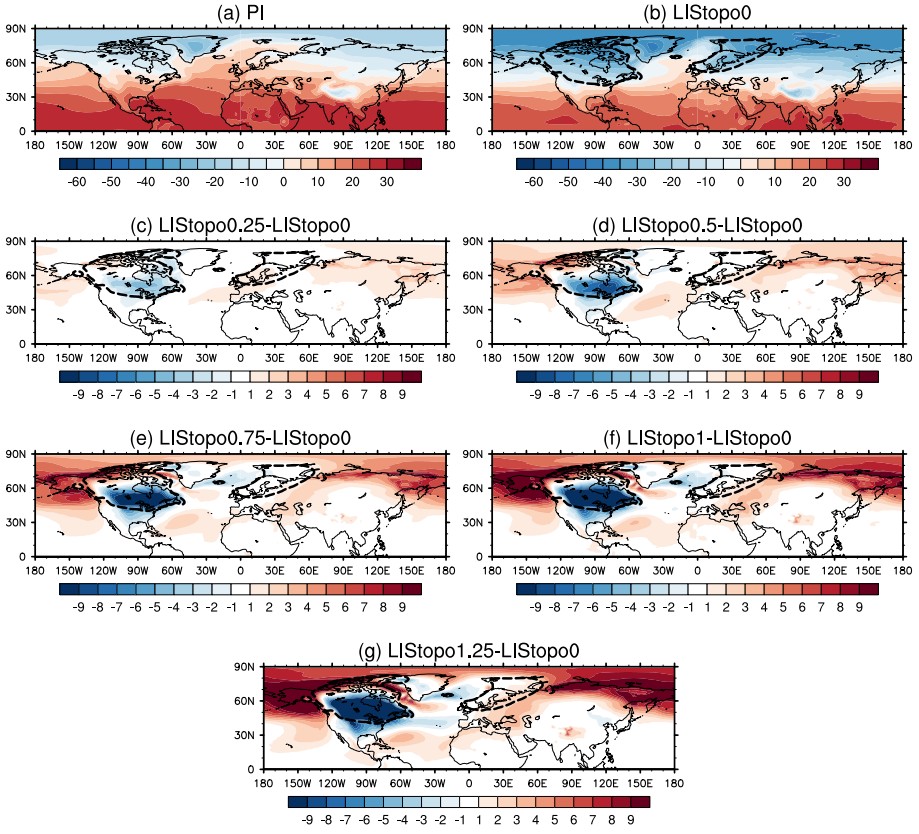

**Figure 1.** Annual-mean surface temperature [°C] in the (a) the PI and (b) the LIStopo0 simulation. Panels c-g show the influence of the LIS elevation on the surface temperature with respect to LIStopo0 in the (c) LIStopo0.25, (d) LIStopo0.5, (e) LIStopo0.75, (f) LIStopo1, and (g) LIStopo1.25 simulations, respectively. The dashed black contours outline the LIS and FIS extents in the LGM simulations.

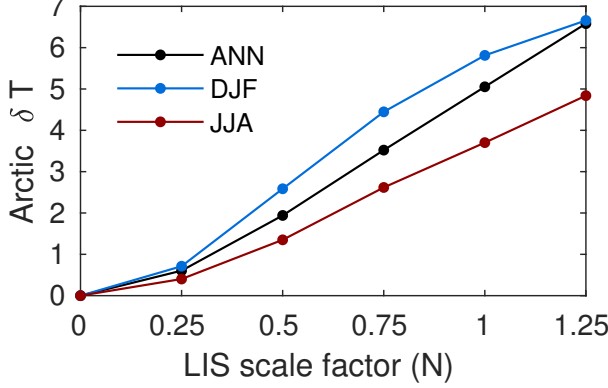

**Figure 2.** The Arctic surface temperature anomaly [°C] (area-weighted average between 70°N and 90°N) with respect to LIStopo0 for the annual-mean (ANN), boreal winter (DJF) and boreal summer (JJA) seasons.



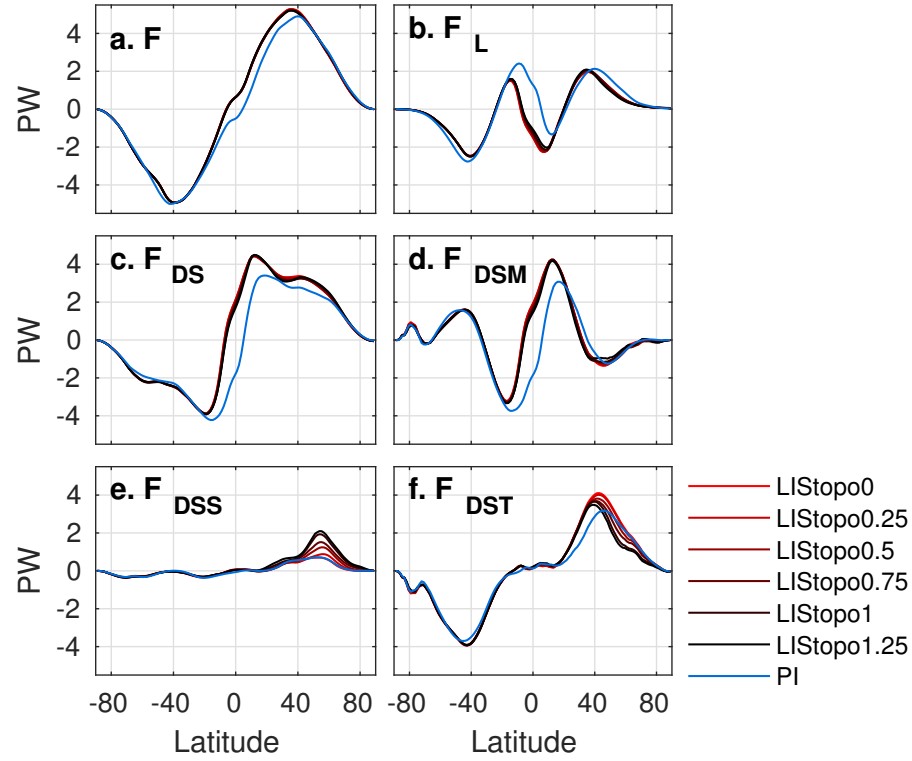

**Figure 3.** Vertically and zonally integrated (annual-mean) atmospheric northward flux [PW= $10^{15}$W] of (a) total energy ($F$), (b) latent energy ($F_L$) and (c) dry-static energy ($F_{DS}$), and dry-static energy contributions from the (d) mean circulation ($F_{DSM}$), (e) stationary eddies ($F_{DSS}$) and (f) transient eddies ($F_{DSS}$).

**Table 1.** The atmospheric flux convergence [W m$^{-2}$] in the Arctic (area-weighted average >70°N) separated into: total energy ($C$), latent energy ($C_L$), dry-static energy ($C_{DS}$). The latter is further decomposed into contributions from the mean circulation ($C_{DSM}$), stationary eddies ($C_{DSS}$) and transient eddies ($C_{DST}$). $\delta C$ shows the change in the total energy-flux convergence with respect to LIStopo0.

|  | $C$ | $C_L$ | $C_{DS}$ | $C_{DSM}$ | $C_{DSS}$ | $C_{DST}$ | $\delta C$ |
|---|---|---|---|---|---|---|---|
| PI | **102** | 17 | 85 | 3 | 9 | 79 | n/a |
| LIStopo0 | **96** | 8 | 88 | 4 | 7 | 84 | n/a |
| LIStopo0.25 | **96** | 8 | 88 | 8 | 13 | 83 | **+0.5** |
| LIStopo0.5 | **97** | 8 | 89 | 12 | 23 | 78 | **+1.8** |
| LIStopo0.75 | **99** | 8 | 91 | 9 | 31 | 69 | **+3.3** |
| LIStopo1 | **101** | 8 | 93 | 2 | 41 | 54 | **+5.0** |
| LIStopo1.25 | **102** | 8 | 94 | 1 | 48 | 47 | **+6.3** |





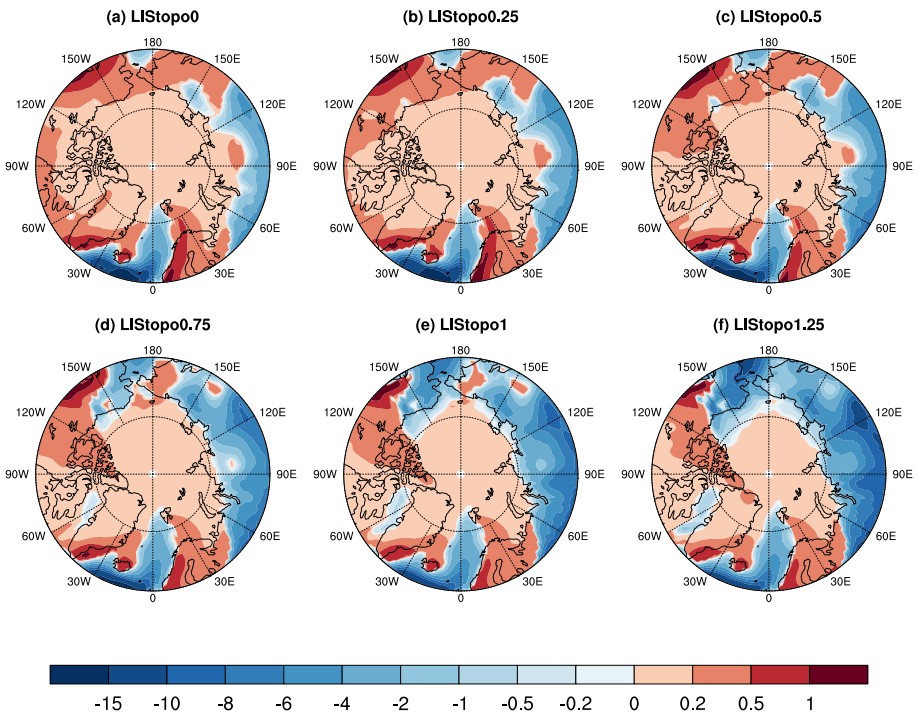

**Figure 4.** Estimated annual SMB [m yr$^{-1}$] in the Arctic region from the LGM simulations.

**Table 2.** Changes in TOA net radiation [W m$^{-2}$] with respect to LIStopo0 ($\delta R$), and estimated contributions to $\delta R$ from changes in surface albedo ($\delta R_\alpha$), water vapor and lapse rate ($\delta R_{wv+lr}$), cloudiness ($\delta R_{cld}$), and surface temperature (Planck feedback; $\delta R_T$).

|  | $\delta R$ | $\delta R_\alpha$ | $\delta R_{wv+lr}$ | $\delta R_{cld}$ | $\delta R_T$ |
|---|---|---|---|---|---|
| LIStopo0.25 | **-0.4** | +0.5 | +0.8 | +0.0 | -1.8 |
| LIStopo0.5 | **-1.7** | +1.1 | +2.6 | +0.4 | -5.8 |
| LIStopo0.75 | **-3.1** | +2.2 | +4.5 | +0.7 | -10.7 |
| LIStopo1 | **-4.8** | +3.4 | +6.6 | +0.6 | -15.7 |
| LIStopo1.25 | **-5.9** | +4.6 | +9.2 | +0.6 | -20.9 |