# Peer review of "Arctic warming induced by the Laurentide ice sheet topography"

_Climate of the Past, 2018_

## Referee Comment (RC1) · D. Ullman (Referee) · 30 Apr 2018

This manuscript presents a new assessment of the influence of Laurentide ice sheet (LIS) height on atmospheric circulation at the last glacial maximum. The experiment design is simple and thus the results are clear. The study involves adjusting the LIS height as a boundary condition in a medium-high resolution atmospheric model coupled to a slab ocean. Each sensitivity simulation is forced by LGM boundary conditions (orbital parameters, greenhouse gases, etc.), with LIS height scaled in 6 separate experiments from a scaling factor of 0 (LIS albedo effect only) to 1.25 (25% taller than LIS reconstructions). This experiment shows that a taller LIS can drive changes in atmospheric circulation that drive widespread warming in the North Hemisphere Arctic, which could serve to be a self-limiting influence on LIS height, through surface mass

balance effects. This work is similar to a variety of earlier publications that perturb LIS height (appropriately referenced by the authors). However, this study is novel in its presentation of a method to disentangle the contributions of meridional energy flux and flux convergence due to mean circulation, stationary eddies, and transient eddies. This separation of atmospheric processes allows the authors to demonstrate that the LIS-height-driven surface warming is dominated by the energy flux from stationary wave eddies, which is mostly compensated by transient eddies. This mechanistic description of the various flux and feedback contributions is particularly useful in understanding how large ice sheets influence climate (as opposed to the other way around). The manuscript is well-written and provides a clear description of the mechanisms controlling atmospheric circulation change due to LIS height. The authors also provide a clear explanation of the study's limitations (particularly the lack of a fully dynamic ocean). I appreciate that the authors have included results from a simple surface mass balance model in their discussion of implications for LIS-height-driven temperature change on the mass balance of the LIS itself. I support this manuscript for publication, but I have a few minor comments that should be considered that may help provide some necessary clarification.

Line 11 (in abstract): "These results suggest a positive feedback between continental-scale ice sheets and the Arctic temperatures that may help constrain LIS elevation..." Why is this a "positive" feedback? I tend to consider positive feedbacks to be amplifying feedbacks. But the mass-balance feedback described in this paper counteracts (or "constrains") the initial change. LIS grows –> warmer Arctic temps –> reduced LIS surface mass balance –> LIS shrinks. Isn't this a NEGATIVE feedback? Please consider changing throughout.

Line 30 → model simulations are 60 years in length; 35y for spinup, 25y for analysis. Is this enough? For the spin-up, can the authors demonstrate with some key atmospheric variables that the simulation is no longer demonstrating drift? Similarly, does 25 years provide enough time to appropriately assess a climatology?

Line 12-17 → The surface mass balance model used in this study is a simple PDD approach. A PDD factor based on observations from modern Greenland might not be completely relevant for the LIS (see Pollard et al., 2000, Global and Planetary Change). It may be worth noting this limitation: that a fully-resolved energy balance model would provide a more complete assessment of surface mass balance. However, Pollard et al. (2000) showed that for paleo applications, conclusions of a PDD approach are generally consistent with an energy balance model. This is to say that I think the general trend of surface mass balance change due to LIS elevation (Fig. 4) is likely robust. However, the observation of positive surface mass balance over Siberia, except in the LIStopo1.25 simulation, might be sensitive to the selection of the PDD factor in the surface mass balance model. Further sensitivity analysis of the PDD factor used in these simulations may be necessary.

––––––––––––––––––––––––––––––––

---

## Referee Comment (RC2) · Anonymous Referee #2 · 7 May 2018

The authors present a study of the impact on the Arctic temperature by varying heights of the Laurentide ice sheets in a AGCM-slab ocean setup under otherwise constant LGM boundary conditions. The experimental setup is clean and allows for direct attribution of effects and very clearly illustrates the role played by the ice sheet height in modifying contributions from standing verses transient eddies to the Arctic energy budget. The study shows that as ice sheet topography increases, the meridional heat (dry static) transport by standing eddies increases enough to overcome the concurrent decrease in transient eddy transport, providing a net increase in meridional heat transport giving rise to a mean Arctic warming. This effect reaches 6.5 degC for an LIS of 125% of the reconstructed LIS, compared to one of 0%.

The paper is well written, organized, referenced and argued. The conclusions are clear

and important and they follow logically from the results presented. I support publication of the manuscript with only a few minor issues to consider in the below.

P3L29: Is the q-flux taken from Liu et al. (2009) also the one used in the PI-experiment? If not, please say so and discuss the impact of this. It shouldn't be important for your conclusions as they follow from comparisons of the various LIStopo experiments. But the PI experiment does enter into Figs 1 and 2 and Table 1, and the interpretation thereof could be influenced by the q-flux used.

P4L4: A little more detail on the construction of the LIStopos is warranted given that they are the centerpiece of the study. In the text it sounds as if you simply multiply the actual elevation by a number, N. But is it rather the anomaly of the LGM topo wrt to PI topo that you scale with N? The fact that the N=0 case corresponds to PI topo tells me that this is rather the case. Otherwise, N=0 would mean completely flat topography.

P4L13-14: Does the qflux change also contribute to the change?

Fig 1: - Consider showing this as in a polar stereographic projection instead. Given that "Arctic" enters into the title of the paper, a highlight of Arctic changes could be in place. - Also, consider showing some standard pressure level height (say Z500) as contours on these plots, to illustrate the stationary eddy changes (if they are visible). The paper talks a lot about the changes in circulation, but nowhere are these changes visualized.

P7L24: Given the importance of this analysis, spend a few sentences outlining the principle in the APRP method.

P8L2: Perhaps add "(not shown)" after the discussion of changes in precipitable water.

P10L16-16: Do you perform the vertical integrals on the time-mean output from the model? This often leads to problems if the output is on (hybrid) sigma levels. Usually this has to be taken care of by performing the vertical integrals on-line on the time-step model state and then outputting time means over the vertically integrated quantities.

How did you do it?

P11L2-4: Could you write a little more on how you arrive at these expressions for the split-up in contributions?

---

## Author Comment (AC1) · 1 Jun 2018

We thank D. Ullman for reviewing the manuscript and providing helpful comments that have improved the manuscript. Please see below for the responses to the specific comments.

**Reviewer comment 1**

*Line 11 (in abstract): "These results suggest a positive feedback between continental-scale ice sheets and the Arctic temperatures that may help constrain LIS elevation..." Why is this a "positive" feedback? I tend to consider positive feedbacks to be amplifying feedbacks. But the mass-balance feedback described in this paper counteracts (or "constrains") the initial change. LIS grows –> warmer Arctic temps –> reduced LIS sur-*

*face mass balance –> LIS shrinks. Isn't this a NEGATIVE feedback? Please consider changing throughout.*

**Reply from authors** Yes, we agree it is a bit confusing. We were thinking of this as a positive feedback in terms of temperature (higher LIS -> more heat transport -> less albedo, more water vapor -> higher temperature -> less albedo, more water vapor ...). You are right that in terms of mass balance it should be a negative feedback. To avoid all confusion, we have removed the word "positive" from the abstract.

**Reviewer comment 2**

*Line 30 -> model simulations are 60 years in length; 35y for spinup, 25y for analysis. Is this enough? For the spin-up, can the authors demonstrate with some key atmospheric variables that the simulation is no longer demonstrating drift? Similarly, does 25 years provide enough time to appropriately assess a climatology?*

**Reply from authors** Model simulation lengths of 60 years is common when using a slab (mixed-layer) ocean model (the same was used in e.g. Bitz et al. 2012; Löfverström et al. 2014). Because there no deep ocean representation in the model, the ocean spin-up is completely determined by the equilibration time-scales of mixed-layer, which is typically 20 years. The relatively short spin-up in our simulations is illustrated in Fig. 1 (enclosed in this document; see below), which shows the time evolution of the (annual) global-mean and Arctic (average north of 70N) surface temperature from the LIStopo0 and LIStopo1.25 simulations. The spin-up phase (as used in the manuscript) is highlighted with dashed lines and the climatological averaging period with solid lines. In Fig. 1, it is evident that steady state is reached after approximately 25 model years.

The reason why we chose to compute the climatology over 25 years instead of say 30 or 35 years is that we noticed that some of the simulations required a few additional years to reach equilibrium. Hence, to be on the "safe side", we decided to compute the climatology over the last 25 years instead of the last 30 or 35 years. However, as is seen in Fig. 1b, the Arctic temperature is not very sensitive to this choice. The

main result of our study, i.e. the LIS-induced Arctic warming (difference between the LIStopo1.25 and LIStopo0 in Fig. 1b), would not change much if the averaging period was 5-10 years longer or shorter than the present choice.

**Reviewer comment 3**

*Line 12-17 -> The surface mass balance model used in this study is a simple PDD approach. A PDD factor based on observations from modern Greenland might not be completely relevant for the LIS (see Pollard et al., 2000, Global and Planetary Change). It may be worth noting this limitation: that a fully-resolved energy balance model would provide a more complete assessment of surface mass balance. However, Pollard et al. (2000) showed that for paleo applications, conclusions of a PDD approach are generally consistent with an energy balance model. This is to say that I think the general trend of surface mass balance change due to LIS elevation (Fig. 4) is likely robust. However, the observation of positive surface mass balance over Siberia, except in the LIStopo1.25 simulation, might be sensitive to the selection of the PDD factor in the surface mass balance model. Further sensitivity analysis of the PDD factor used in these simulations may be necessary.*

**Reply from authors** Good idea. We have added a comment about the uncertainty of PDD models in the discussion section.

**References**

Bitz, C. M., Shell, K., Gent, P., Bailey, D., Danabasoglu, G., Armour, K., Holland, M., and Kiehl, J.: Climate sensitivity of the community climate system model, version 4, J. Climate, 25, 3053–3070, 2012.

Löfverström, M., Caballero, R., Nilsson, J., and Kleman, J.: Evolution of the large-scale atmospheric circulation in response to changing ice sheets over the last glacial cycle, Climate of the Past, 10, 1453–1471, https://doi.org/10.5194/cp-10-1453-2014, 2014.

[Figure]

**Fig. 1.** Temporal evolution of the global-mean (a) and Arctic (b) annual-mean surface temperature in the LIStopo0 (red lines) and LIStopo1.25 (black lines) simulations.

---

## Author Comment (AC2) · 1 Jun 2018

We thank Anonymous Referee 2 for the insightful comments on the manuscript. Please see below for the responses to the specific comments.

**Reviewer comment 1**

P3L29: Is the q-flux taken from Liu et al. (2009) also the one used in the PI-experiment? If not, please say so and discuss the impact of this. It shouldn't be important for your conclusions as they follow from comparisons of the various LIStopo experiments. But the PI experiment does enter into Figs 1 and 2 and Table 1, and the interpretation thereof could be influenced by the q-flux used.

Reply from authors Thanks for noticing this. The q-flux in the PI experiment is derived

from the surface energy balance in an atmospheric model experiment with (prescribed) observed sea-surface conditions, PI insolation, and PI greenhouse gas concentrations (same as in Löfverström et al. 2014 and Liakka et al. 2016). We have clarified this in section 2 of the revised manuscript.

**Reviewer comment 2**

P4L4: A little more detail on the construction of the LIStopos is warranted given that they are the centerpiece of the study. In the text it sounds as if you simply multiply the actual elevation by a number, N. But is it rather the anomaly of the LGM topo wrt to PI topo that you scale with N? The fact that the N=0 case corresponds to PI topo tells me that this is rather the case. Otherwise, N=0 would mean completely flat topography.

**Reply from authors** Yes, the LIS topography scaling factor (N) has been applied to the LIS topography, which is evaluated as the difference from the PI topography. Therefore, N=0 corresponds to the PI topography in North America with the albedo from the LGM LIS. We have clarified this in section 2.

**Reviewer comment 3**

P4L13-14: Does the qflux change also contribute to the change?

**Reply from authors** Yes, potentially it does. We have highlighted that the q-flux is a potential candidate for explaining the differences between LIStopo0 and PI in section 3.1.

**Reviewer comment 4**

Fig 1: - Consider showing this as in a polar stereographic projection instead. Given that "Arctic" enters into the title of the paper, a highlight of Arctic changes could be in place. - Also, consider showing some standard pressure level height (say Z500) as contours on these plots, to illustrate the stationary eddy changes (if they are visible). The paper talks a lot about the changes in circulation, but nowhere are these changes visualized.

CPD
**Reply from authors** This is a very good idea. We have updated the figure so that it now also includes the eddy Z500 field and is shown in polar stereographic projection.

**Reviewer comment 5**

*P7L24:* Given the importance of this analysis, spend a few sentences outlining the principle in the APRP method.

**Reply from authors** We have added a sentence which briefly explains the essentials of the APRP method in section 3.3 of the manuscript.

**Reviewer comment 6**

P8L2: Perhaps add "(not shown)" after the discussion of changes in precipitable water.

Reply from authors Thanks. We have added that.

**Reviewer comment 7**

P10L16-16: Do you perform the vertical integrals on the time-mean output from the model? This often leads to problems if the output is on (hybrid) sigma levels. Usually this has to be taken care of by performing the vertical integrals on-line on the time-step model state and then outputting time means over the vertically integrated quantities. How did you do it?

**Reply from authors** The integrals were computed on pressure levels. The heat flux quantities were first interpolated from the 26 hybrid levels to 20 equally spaced pressure levels, ranging from 25 hPa to 975 hPa. In the numerical integration, each pressure level then represents the mid-point pressure of a 50 hPa thick pressure layer. The integration for each vertical column was carried out from the top of the atmosphere to the surface, which is represented by the climatological monthly-mean surface pressure. We have added some explanatory sentences about this to Appendix A.

**Reviewer comment 8**

**P11L2-4: Could you write a little more on how you arrive at these expressions for the split-up in contributions?**

Reply from authors We arrived at those expressions from the following observations:

- The surface temperature contributions to the net outgoing LW change between two simulations is simply the difference between the outgoing LW from the surface between those simulations (Eq. B1 in the manuscript).
- The cloud contributions are estimated to be the difference between the total and clear-sky (i.e. non-cloud) LW changes (Eq. B3).
- The contributions from water vapor and lapse rate are assumed to be equal to the LW change in the clear-sky variables minus the LW change at the surface (Eq. B2). The reason why the surface LW needs to be subtracted is to account for lapse-rate changes (determined by the difference in LW between the TOA and surface).

Note that any change in other radiative forcing agents (e.g. CO2 and aerosols) would also influence Eq. B2. However, because the LIStopo simulations use identical aerosol datasets (representative for PI) and have the same concentrations of CO2 and other greenhouse gases, only changes in water vapor and lapse rate are important for Eq. B2 in our case.

The accuracy of the individual contributions is (at least partly) validated by the fact that summing up Eqs. B1 to B3 yields the total LW change at TOA. Hence, an alternative way to understand the split-up equations is to first subtract the (relatively straightforward) surface temperature and cloud contributions from the total LW change. The residual term then contains "everything else", which in our case reflects changes in water vapor and lapse rate, as the other radiative forcing agents are the same in all LGM (LIStopo) experiments.

CPD
We have added some of this discussion to Appendix B.

**References**

Löfverström, M., Caballero, R., Nilsson, J., and Kleman, J.: Evolution of the large-scale atmospheric circulation in response to changing ice sheets over the last glacial cycle, Climate of the Past, 10, 1453–1471, https://doi.org/10.5194/cp-10-1453-2014, 2014.

Liakka, J., Löfverström, M., and Colleoni, F.: The impact of the North American glacial topography on the evolution of the Eurasian ice sheet over the last glacial cycle, Climate of the Past, 12, 1225–1241, https://doi.org/10.5194/cp-12-1225-2016, 2016.